# Harmful Effect of Intrauterine Smoke Exposure on Neuronal Control of “Fetal Breathing System” in Stillbirths

**DOI:** 10.3390/ijerph19074164

**Published:** 2022-03-31

**Authors:** Anna M. Lavezzi, Teresa Pusiol, Beatrice Paradiso

**Affiliations:** 1“Lino Rossi” Research Center for the Study and Prevention of Unexpected Perinatal Death and SIDS, Department of Biomedical, Surgical and Dental Sciences, University of Milan, 20121 Milan, Italy; beatrice.paradiso@unife.it; 2Institute of Anatomic Pathology, APSS, 38122 Trento, Italy; teresa.pusiol@apss.tn.it; 3General Pathology Unit, Dolo Hospital, 30031 Dolo, Italy

**Keywords:** cigarette smoke, pregnancy, SIUDS, neuropathology, brainstem, lung development, toxic nanoparticles

## Abstract

This article is aimed to contribute to the current knowledge on the role of toxic substances such as nicotine on sudden intrauterine unexplained deaths’ (SIUDS’) pathogenetic mechanisms. The in-depth histopathological examination of the autonomic nervous system in wide groups of victims of SIUDS (47 cases) and controls (20 cases), with both smoking and no-smoking mothers, highlighted the frequent presence of the hypodevelopment of brainstem structures checking the vital functions. In particular, the hypoplasia of the pontine parafacial nucleus together with hypoplastic lungs for gestational age were observed in SIUDS cases with mothers who smoked cigarettes, including electronic ones. The results allow us to assume that the products of cigarette smoke during pregnancy can easily cross the placental barrier, thus entering the fetal circulation and damaging the most sensitive organs, such as lungs and brain. In a non-negligible percentage of SIUDS, the mothers did not smoke. Furthermore, based on previous and ongoing studies conducted through analytical procedures and the use of scanning electron microscopy, the authors envisage the involvement of toxic nanoparticles (such as agricultural pesticides and nanomaterials increasingly used in biomedicine, bioscience and biotechnology) in the death pathogenesis, with similar mechanisms to those of nicotine.

## 1. Introduction

Breathing in humans is mainly regulated by a network of neurons located in the brainstem/rostral spinal cord [1]. More specifically, the ventilatory rate and the duration of the breathing phases depend on the balance between stimulatory and inhibitory neurons, which interconnect to one another to form a “respiratory network (RN) [2]. The RN is able to increase or decrease ventilation in accordance with changes in carbon dioxide production and oxygen consumption. For this purpose, a large alveolar surface is required in the lungs to allow sufficient oxygen diffusion. Hence, pulmonary organogenesis must primarily be aimed at expanding the airways and gas exchange surface. Although, during intrauterine life, the fetus relies on the placenta for oxygenation rather than the lungs, fetal breathing (FB) movements occur to promote pulmonary development and respiratory muscle maturation [3].

Therefore, FB is regulated differently than postnatal respiration by the RN [2]. FB is detectable from as early as 10 weeks of gestation, and it varies throughout pregnancy, with a progressive increase in the amplitude and frequency of respiratory movements, according to the directives of the RN [4,5,6,7,8]. Defective development and/or malfunction of one or more RN components can cause prolonged apneic periods between successive breathing episodes in utero and, even more, a total absence of FB, resulting in incomplete airway development with consequent pulmonary hypoplasia [9,10].

An in-depth study on the neural regulation of lung development and respiratory rhythm in utero, which we define as the “fetal breathing system” (FBS), can provide us with a better understanding of the pathogenetic mechanisms underlying impaired pulmonary development, a condition frequently found in unexplained stillbirths.

The subjects used for this study were collected in accordance with the Italian Law 31/2006 “Regulations for Diagnostic Post Mortem Investigation in Victims of Sudden Infant Death Syndrome (SIDS) and Unexpected Fetal Death (SIUDS: Sudden Intrauterine Unexplained Death Syndrome)” [11]. This law states, in particular, that all stillborn babies that die without any apparent cause after 25 weeks of gestation must be submitted to an in-depth diagnostic postmortem examination, including a detailed analysis of the brain centers, checking the vital functions and investigation on the risk factors [12,13].

Firstly, a group of SIUDS was selected for which the autopsy investigations highlighted hypoplastic lungs for gestational age as the only pathological finding. Then, in these cases, the possible presence of developmental alterations of the brain centers involved in breathing control and any risk factors that could have significant effects on the development of the FBS were investigated. The results were compared with those obtained from a group of age-matched unexplained stillbirths with normal lung development and a control group composed of fetuses who died of a well-established illness.

The main objectives of this study were (1) to evaluate in each case the main neuronal vital centers of the brainstem and the lung developmental phase with respect to the gestational age and (2) to correlate the results with the identified risk factors.

## 2. Material and Methods

Sixty-seven cases (25–40 gestational weeks; 33 males and 34 females) were selected from a wide group of fresh ante-partum stillbirths, in accordance with the aforementioned Italian law and for which extensive anamnestic information was available. Comprehensive fetal postmortem examinations of all cases, including umbilical cord/placental/membrane evaluation and a careful analysis of the pulmonary developmental stage, were performed by expert pathologists. On the basis of their diagnoses, the stillbirths were classified into three groups: (1) SIUDS, in which the routine postmortem examination did not establish the exact cause of death but only a delayed pulmonary development for gestational week, a pathological finding not incompatible with prenatal life (25 cases); (2) SIUDS cases with normal lung development for gestational age (22 cases) and (3) stillbirths for which the cause of death was determined at autopsy and used as controls (20 cases). SIUDS cases and controls were similar based on matched factors, i.e., gestational age, sex and weight at death (the latter is understood to be within 100 g, as it was impossible to match birthweights exactly).

Information for each case was collected through a specific questionnaire on the main risk factors associated with stillbirths, such as maternal age; ethnicity; socioeconomic status; obesity; cigarette (including electronic cigarette) smoking; alcohol and drug abuse and, whenever detectable, occupational and/or residential exposure to high levels of airborne particulate matter, pesticides, solvents and other toxicants. Regarding maternal smoking habits, in order to validate the mothers’ smoking statements, especially in cases of denial, locks of each victim’s hair (or their mother’s in case of unavailability) were removed to test for cotinine, which is the main nicotine metabolite characterized by a long half-life and great stability [14]. None of the mothers suffered from any significant illness, including diabetes and hypertensive disorders (such as chronic hypertension and preeclampsia). 

### 2.1. Anatomopathological Procedures

Samples of major organs (heart, lungs, liver, kidneys and brain) taken at routine autopsy were fixed in 10% phosphate-buffered formalin, processed and embedded in paraffin. Five-μm-thick sections were cut and stained with hematoxylin–eosin and Heidenhain’s trichrome (Azan).

### 2.2. In-Depth Specific Protocols

(1)Brainstem examination: Consecutive brainstem samples taken from the third lower portion of the midbrain to the initial part of the spinal cord, after being processed according to the aforementioned histological procedures, were cut serially. The main nuclei in targeted histological sections stained with Klüver/Barrera, which is the dye of choice for the analysis of nervous tissues, were then analyzed, i.e., the superior olivary complex; the locus coeruleus; the Kölliker–Fuse nucleus; the parafacial nucleus and the median, magnus, dorsal, caudal and linear raphe nuclei in the sections obtained from the pontine samples and the hypoglossus, the dorsal motor vagus, the tractus solitarius, the ambiguus, the pre-Bötzinger, the inferior olivary, the arcuate, the obscurus and pallidus raphe nuclei in the medulla oblongata sections and the intermediolateral nucleus in the sections from the more caudal sample (rostral spinal cord).(2)Lung examination: Pulmonary hypoplasia was diagnosed first macroscopically, on the basis of the correlation between lung and body weight (LW/BW), and then according to specific microscopic criteria applied to histological sections taken parallel to the frontal plane and passing through the hilum, which were used for this analysis, i.e., (1) the presence of cartilaginous bronchi up to the distal peripheral level and (2) the radial alveolar count (RAC). This last parameter was obtained by examining at least 10 microscopic fields for each case (with a 40× magnification lens) in order to estimate the number of alveoli transected by a perpendicular line drawn from the center of the most peripheral bronchiole (recognizable by not being completely covered by epithelium) to the pleura or the nearest interlobular septum (15). Table 1 shows the reference values for the diagnosis of prenatal pulmonary hypoplasia.

### 2.3. Statistical Analysis

The statistical significance of direct comparisons between the groups of victims was determined using the analysis of variance (ANOVA). Statistical calculations were carried out with SPSS statistical software (version 11.0; SPSS Inc., Chicago, IL, USA). The differences were regarded as statistically significant if *p* values were <0.05.

## 3. Results

### 3.1. SIUDS

Gross and histological examination of the placental disk, umbilical cord, membranes and all the organs did not reveal any pathology which could have caused the death.

(a)SIUDS group with lung hypoplasia

Twenty-five SIUDS victims were diagnosed with bilateral pulmonary hypoplasia, as they had LW/BW values below 0.012, showing at microscopic examination the presence of peripheric cartilaginous bronchi with few and small peripheral airspaces and, above all, RAC indices lower than the reference values reported in Table 1. Endoalveolar and endobronchial cornea scales were frequently observed in these cases. The microscopic examination of histological sections of the brainstem highlighted developmental alterations of several neuronal centers. The hypoplasia of different brainstem nuclei, more specifically of the parafacial nucleus, the pre-Bötzinger nucleus, the Kölliker–Fuse nucleus, the raphe obscurus and the intermediolateral nucleus, were the main findings. It is important to note that hypoplasia of the parafacial nucleus (pFn) was a common feature in all SIUDS cases of this group. Figure 1 shows the histology of pFn hypoplasia and pulmonary hypoplasia, jointly present in one of these cases.

(b)SIUDS group with normal lung development

Normal pulmonary development was observed in 22 SIUDS cases, as they had normal LW/BW ratios (greater than 0.012), no lobulation anomalies and RAC values in agreement with those shown in Table 1. The neuropathological examination of the brainstem revealed that the most frequent findings were hypoplasia of the Kölliker–Fuse, the pre-Bötzinger, the parafacial and the raphe obscurus nuclei.

### 3.2. Control Group

The well-defined death causes of the 20 cases used as controls were divided into two categories: (1) abnormalities of the placenta, membranes and umbilical cord and (2) fetal pathologies (cardiomyopathies, severe renal dysplasia, pneumonia, septicemia, karyotype abnormalities). In five cases, the neuropathological examination revealed neuronal alterations in the medulla oblongata and, precisely, hypoplasia of the raphe nucleus and of the arcuate nucleus, which are defects that are mostly deemed to be compatible with life [16].

Table 2 summarizes the neuropathological findings of the three groups (two SIUDS and control group).

### 3.3. Distribution of Risk Factors among the Three Groups

Overall, 34 of the 47 SIUDS cases (72%) and 4 of the 20 control cases (20%) were exposed to tobacco smoke during the whole pregnancy and often even before the onset of pregnancy, consuming more than 3 cigarettes/day, based on maternal self-reported smoking or cotinine tests. It should be noted that 24 of the 34 SIUDS exposed to cigarette smoke in utero belonged to the group with pulmonary hypoplasia. Among the smoking mothers in this latter group, 5 routinely smoked electronic cigarettes (e-cigs) and a further 4 used both e-cigs and traditional cigarettes.

The remaining 29 mothers had no history of cigarette smoking, verified using the same criteria. No significant differences were observed regarding the other risk factors, such as gender, ethnicity, maternal age, overweight, socioeconomic status and drug and/or alcohol abuse. Lastly, it was impossible to evaluate a significant role of exposure to environmental toxicants (particulate matter, pesticides, polycyclic aromatic hydrocarbons, toxic metals, etc.), since these risk factors were ascertained or considered to be highly causative only in 12 SIUDS (26%) and 2 control (10%) cases whose mothers reside or work in agricultural areas where pesticides are often used or close to industries utilizing toxic nanomaterials. Table 3 summarizes the characteristics of the case study, including the risk factors.

### 3.4. Correlation between Neuropathological Findings and Maternal Smoking in Pregnancy

Table 4 shows the relationship between in-utero tobacco exposure, which was the only risk factor significantly related to SIUDS found in this study, and the main neuropathological findings in the brainstems. A very high correlation was found between maternal smoking, pulmonary hypoplasia, pFn hypoplasia and SIUDS cases.

## 4. Discussion

It is widely known that maternal tobacco smoking during pregnancy is significantly associated with fetal and neonatal morbidities (such as placental abnormalities, fetal growth restriction, birth complications, preterm birth, low birthweight and congenital malformations) and even mortality, including unexplained perinatal deaths (in particular, SIUDS and SIDS) [17,18,19,20,21,22]. However, the underlying physiological mechanisms of the harmful effects of cigarette smoking are not currently quite understood. Therefore, updates on these issues, particularly those relating to fetal death, are highly desirable.

This study has contributed to the current knowledge on this topic by highlighting a specific relationship between intrauterine tobacco smoke exposure and delayed maturation of fetal lungs, a condition that is known to be associated with severe respiratory distress after birth [23,24,25,26]. We believe, however, that defective prenatal respiratory movements can also be responsible for unfavorable outcomes before birth. In fact, our results indicate that maternal cigarette smoking during pregnancy may have adversely affected the “fetal breathing system” (FBS), a term that encompasses the development of fetal lungs, respiratory activity in utero and their control by specific nerve centers of the RN in sudden unexplained fetal deaths.

It is well known that although they are not involved in fetal oxygenation, fetal breathing movements play an essential role in lung and respiratory muscles’ development [3,4,5,6,7,8]. Therefore, nicotine, the most harmful chemical contained in tobacco smoke, can be considered to be a major risk factor highly involved in the pathogenesis of pulmonary hypoplasia in stillbirths. Our current understanding of the sequence of pathophysiological mechanisms that may have contributed to these adverse effects is as follows: when a mother smokes in pregnancy, the smoke is rapidly absorbed by the lung airways and alveoli and enters the maternal bloodstream. Due to the optimal hematic pH (approximately 7.4%), nicotine is able to pass through cell membranes [27], a property that allows it to easily cross the placental barrier and enter into the fetal O_2_ circulation, where it can be detected at levels exceeding maternal concentrations by 15% [28,29]. Then, nicotine, being one of the few fat-soluble substances capable of crossing the blood–brain barrier (BBB), easily enters the brain parenchyma [30], where it can interfere with the development of the nerve centers that control intrauterine respiratory activity and, therefore, also pulmonary growth. This can be facilitated by cerebral hypoxia induced by tobacco carbon monoxide (CO), a gaseous nicotine combustion, which is similarly capable of crossing the placenta by passive diffusion and entering into the fetal blood system. Here, the CO, having a greater affinity to hemoglobin than oxygen, increases the levels of carboxyhemoglobin, which is a stable complex that inhibits the release of O_2_ into fetal tissues [31]. This hypoxic condition affects especially the most susceptible organs like the lungs [32] and, above all, the brain, where oxygen deficiency can alter the expression of genes checking the nervous system development [33,34]. Furthermore, it must be considered that the placenta is permeable even to cadmium oxide, which is another compound generated by tobacco smoking [35]. Once it has entered the fetal blood circulation, cadmium has the ability to directly accumulate in lung tissues, impairing their structure and functionality, and also to alter the cell–cell junctions in the endothelium of the BBB with consequent neurotoxic effects [36,37,38,39].

The association between maternal smoking in pregnancy, pontine pFn hypoplasia and lung hypoplasia observed in most SIUDS cases with smoking mothers supports our assumption that the pulmonary hypoplasia may be caused by a specific defective neuronal development triggered by cigarette smoke exposure. We must then consider that intrauterine breathing is not only aimed at the development of the lungs, but also at the modulation of the vagal control of the fetal heart. Precisely, the cardiovagal activity, resulting from the parasympathetic innervation of the heart mediated by the vagus nerve, under normal conditions is reduced during inspiration and increased during expiration. This cardiorespiratory coupling (also named “Respiratory sinus arrhythmia”) can be considered a physiological unified vital rhythm which, under the control of specific neural centers, is able to coordinate respiratory and sympathetic responses to hypoxia [40,41,42,43].

Therefore, we can assume that immaturity in cardiorespiratory coupling, resulting from the action of tobacco on pFn development, can impair responses to cardiorespiratory stress and increase the sudden fetal death risk. In this regard, experimental studies have shown that the pFn is just as effective in breathing control, although its exact role in this activity has not yet been well defined. In studies conducted on mouse brain transections, Janczewski and Feldman [44] reported the existence of two functionally distinct rhythm generators in the brainstem, precisely the pre-Bötzinger nucleus (preBötn) for inspiration and the pFn for expiration. Subsequent animal studies of Onimaru et al. [45,46] indicated that the pFn is predominantly composed of pre-inspiratory neurons merely aimed at stimulating the inspiratory center and that this coupling (pFn/preBötn) is especially important in the generation of the primary respiratory rhythm at birth. More recently, the pFn has been defined as a conditional expiratory oscillator during eupnea, responsible for the inspiratory/expiratory phase transition control [47,48,49]. Furthermore, according to these studies, its activation is required, above all, under conditions of elevated metabolic demand, such as in cases of hypoxia, in order to increase the active exhalation to support the required pulmonary hyperventilation, thus safeguarding life [50].

The frequent finding of pFn hypoplasia in SIUDS and, notably, its constant presence in cases with pulmonary hypodevelopment and smoking mothers demonstrates not only that this nucleus is active even before birth, but also that its normal structure and functionality is indispensable for a regular conclusion of the pregnancy, especially if there is oxygen deficiency, such as that caused by the fetal smoke absorption.

Our results have provided new insight into the well-known association between smoking during pregnancy and stillbirth, having highlighted for the first time, to the best of our knowledge, the adverse effect of nicotine on the intrauterine pulmonary development through targeted action on a specific respiratory neuronal center. This certainly also applies to the nine cases in which mothers smoked e-cigs in pregnancy and with hypodevelopment of both the pFn and lungs.

E-cigs belong to the latest forms of tobacco products widely used in the last decade, even in pregnancy, as they are considered a safer alternative to traditional cigarettes. However, it has been shown in the past few years that the aerosol produced by e-cigs contains a similar and even greater amount of nicotine than conventional tobacco cigarettes, as well as other harmful products, including formaldehyde, traces of metals and small particles [51,52]. Animal studies have documented the detrimental effects of gestational exposure to tobacco products on the developing fetus, including the alteration or delay of fetal lung growth [53,54,55].

In relation to the SIUDS cases with a non-smoking mother reported in this study, we can assume that other toxic factors may have interfered directly with the development of vital nerve centers. Maternal exposure during pregnancy to nanosized environmental pollutants (NEPs), such as pesticides, solvents and varnishes, have proved in fact to be positively associated with adverse intrauterine effects, leading to fetal growth retardation, preterm birth and stillbirth [56,57,58].

Our previous articles showed the presence of neurodevelopmental disorders in sudden intrauterine deaths related to pesticide exposure, highlighting, in particular, the impact of organochlorine and organophosphate insecticides used in agriculture on neuronal α7-nicotinic acetylcholine receptor expression in brainstems [59,60].

The recent finding in our research, currently in progress, of metallic debris in the brain of five of the SIUDS cases of the present study with normal lung development validates the assumption of NEPs’ involvement in the sudden fetal death. More specifically, inorganic nanoparticles (composed primarily of Iron, Silver and Aluminum), probably derived from nanomaterials increasingly used in biomedicine, bioscience and environmental biotechnology, have been detected in the brainstem tissue samples by using Scanning Electron Microscopy (SEM) techniques. It is possible to speculate that the NEPs can sequentially: (1) enter the bloodstream of a pregnant woman through inhalation, ingestion, injection or skin contact; (2) easily cross like nicotine the placenta; (3) enter into the fetal bloodstream and (4) go beyond the fetal BBB [58,61]. Then, NPs may interfere with normal brain development and consequently participate in the pathogenic mechanism of death [62,63,64]. Given the highly encouraging data obtained, we aim to extend our knowledge by examining through SEM all the cases of this study to validate the harmful role impacting the fetus of atmospheric pollution. We propose that since pulmonary hypoplasia is not only associated with high intrauterine mortality, but also that, if gestation comes at the end, it may lead to severe respiratory distress after birth and to neonatal death, a survey on the fetal lung maturity in relation to the gestational age (by measuring the level of surfactant in the amniotic fluid or, even better, through MRI in all the pregnancies [65,66,67], even more so where the mother is found to be a smoker or lives in a proven highly polluted environment) is highly advisable.

## 5. Concluding Remarks

Although much is known about the harmful effects of smoking during pregnancy, maternal smoking rates are decreasing very slowly, thus remaining a serious threat to global health. Thus, it is essential, also on the basis of the results here presented, to accelerate the implementation of informative campaigns and preventive strategies with the aim of helping women to stop tobacco smoking and also e-cigs, whose use continues to increase (especially among younger generations), both prior to conception and during pregnancy. Furthermore, it could be hopeful to provide a better and more reliable explanation of the SIUDS’ pathogenetic mechanism, also considering the detrimental effect of nanosized pollutants on fetal nervous system development. Then, all pregnant women must be made aware of the risk factors that can endanger their baby and avoid not only the smoking habit, but also, as much as possible, at least decrease the exposure to pollutants.

## Figures and Tables

**Figure 1 ijerph-19-04164-f001:**
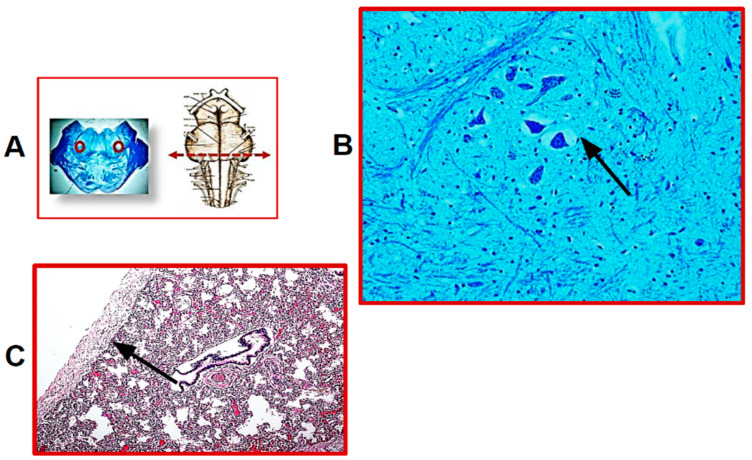
(**A**) On the left: representative histological section of the caudal pons showing, in the circled areas, the localization of the parafacial nucleus; on the right: indication of the level of the histological section. (**B**) Hypoplasia of the parafacial nucleus with few neurons in a SIUDS case (female, 39 gestational weeks). (**C**) Histological section showing lung hypoplasia in the same SIUDS case. The black line indicates the RAC (radial alveolar count). (**A**,**left**) Klüver–Barrera staining, magnification 0.5×; (**B**) Klüver–Barrera staining, magnification 20×; (**C**) Hematoxylin/eosin staining, magnification 10×.

**Table 1 ijerph-19-04164-t001:** Diagnosis of fetal pulmonary hypoplasia ^1^.

**Macroscopic Criteria**
• Lung Weight/Body Weight (LW/BW) < 0.012
**Microscopic criteria**
• Presence of cartilaginous bronchi at the distal level
• RAC * (radial alveolar count)
24–27 gw: <2.2
28–31 gw: <2.6
32–35 gw: <3.2
36–39 gw: <3.6
40 gw: <4.4

***** number of alveoli cut by an imaginary line drawn from a terminal bronchiole to the pleura (or nearest interlobular septum). gw = gestational week. **^1^** adapted from Askenazi, S.S.; Perlman, M. Pulmonary hypoplasia: lung weight and radial alveolar count as criteria of diagnosis. Arch. Dis. Child. 1979, 54, 614–61 [15].

**Table 2 ijerph-19-04164-t002:** Neuropathological findings in SIUDS and Control cases.

Cases		Main Neuropathological Brainstem Findings
SIUDS (47)	with lung hypoplasia (25)	parafacial nucleus hypoplasia (12)
		parafacial nucleus together with pre-Bötzinger nucleus hypoplasia (8)
		parafacial nucleus together with obscurus raphe nucleus hypoplasia (3)
		parafacial nucleus together with pre-Bötzinger nucleus, intermediolateral nucleus and raphe obscurus nucleus hypoplasia (2)
	with normal lung development (22)	Kölliker–Fuse nucleus hypoplasia (7)
		pre-Bötzinger nucleus hypoplasia (6)
		parafacial nucleus hypoplasia (5)
		Kölliker–Fuse nucleus together with parafacial nucleus and pre-Bötzinger nucleus hypoplasia (4)
CONTROLS (20)		no alterations (15)
		arcuate nucleus together with raphe obscurus nucleus hypoplasia (5) *

Categorical data are expressed as the number of cases (in brackets); SIUDS = Sudden Intrauterine Unexplained Death Syndrome. * These are slight alterations of the medulla oblongata that are compatible with life.

**Table 3 ijerph-19-04164-t003:** Case features of the study.

	Stillbirths with Unknown Death Cause (SIUDS) 47 Cases		Stillbirths with Known Death Cause ^1^ (Controls) 20 Cases
	Lung Hypoplasia	Normal Lung Development	
**number of cases**	25	22	20
**gestational weeks**			
range	26–40	25–40	27–39
**fetal weight**			
small for gestational age	13	11	11
normal for gestational age	12	11	9
**fetal sex**			
male	11	12	10
female	14	10	10
**maternal age**			
<20 years	11	10	9
≥20 years	14	12	11
**maternal ethnicity**			
white/black/asian	18/0/7	16/1/5	15/0/5
**maternal socioeconomic status**			
low	13	11	11
normal/high	12	11	9
**maternal overweight**			
yes	6	4	8
no	19	18	12
**maternal smoking ****			
yes	24 ^2^ (96%)	10 (45%)	4 (20%)
no	1	12	16 (80%)
**maternal drug and/or alcohol consumption**			
yes	2	3	4
no	23	19	16
**occupational/environmental toxicant exposure**			
ascertained	2	10	2
unknown	20	10	18

The table shows the distribution of the sample proportions. ^1^ placenta, cord and membrane abnormalities; cardiomyopathies; pneumonia; renal dysplasia; septicemia; karyotype anomalies. ^2^ Nine mothers routinely smoked e-cigs (four of which smoked both e-cigs and traditional cigarettes). ** Statistical significance of SIUDS vs. controls: *p* < 0.01; SIUDS = Sudden Intrauterine Unexplained Death Syndrome.

**Table 4 ijerph-19-04164-t004:** Correlation of the main neuropathological findings with maternal smoking in SIUDS and Control cases.

Cases	Main Neuropathological Brainstem Findings	Maternal Smoking	
		Yes	No
SIUDS (47)		34 (72%)	13 (28%)
with lung hypoplasia (25)	parafacial respiratory nucleus hypoplasia (25 cases)	24 **	1
	pre-Bötzinger nucleus hypoplasia (10 cases)	10 **	0
with normal lung development (22)	Kölliker–Fuse nucleus hypoplasia (11)	6	3
	pre-Bötzinger nucleus hypoplasia (10)	5	5
CONTROLS (20)	arcuate nucleus hypoplasia (5)	0 ^1^	5

The Table shows the distribution of the sample proportions. ****** Statistical significance of SIUDS with lung and parafacial respiratory nucleus/pre-Bötzinger nucleus hypoplasia in relation to maternal smoking: *p* < 0.01. ^1^ Note the association between hypoplasia of the arcuate nucleus (an alteration without particular pathological significance) and the absence of smoking absorption in pregnancy in control cases.

## Data Availability

The data presented in this study are available in anonymized form upon request from the corresponding author.

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
