# Peer review of "Harmful Effect of Intrauterine Smoke Exposure on Neuronal Control of “Fetal Breathing System” in Stillbirths"

_ijerph, 2022, doi:10.3390/ijerph19074164_

Round 1

Reviewer 1 Report

This article investigates the association between the presence or absence of lung hypoplasia and the presence or absence of congenital anomalies of the central respiratory system concerning fetal tobacco exposure. The autopsy is highly reliable, and the high reliability of the specimens compensates for the small number of cases.

Theoretically, the association is convincing and, with minor modifications, deserves to be published.

(Minor Problems)

1) L-120: Table 1 is inferred to be taken from reference 15. References cited should be in the title or footnotes of Table 1.

2) L79-82, Table 2L162-164: Five cases with congenital anomalies were included in the control cases. Since autopsy was used, congenital anomalies can be a confounding factor in SIUDS and should be excluded. Cases that do not include congenital components, such as cases that died of apparent perinatal problems, should be used as control cases.

3) There is a problem with the description of statistical methods. The method section (L125-128) clearly states only ANOVA, but Tables 3 and 4 are statistical studies of proportions. It is necessary to specify the test method in the footnotes of Tables 3 and 4, rather than just stating that SPSS was used.

4) In Table 4, it is stated that there was a significant difference only in the top row, 24 with lung hypoplasia (**), but why is there no significant difference in the bottom row, 10-0, and the bottom row, 0-5? Since a 0 in the ratio calculation usually always results in a significant difference, it is necessary to clearly state this in the paper together with the statistical method in 3), and the reason why no significant difference was found should be stated in the discussion.

5) Limitation is not mentioned in the discussion, but is it not a limitation that the number and amount of smoking were not investigated (or at least not mentioned in the paper)? It is hard to believe that the risk of smoking even one cigarette during the entire gestational period is equivalent to smoking more during the entire gestational period.

Author Response

Point 1:  L-120: Table 1 is inferred to be taken from reference 15. References cited should be in the title or footnotes of Table 1.

Response 1: thank you for this note. We added the reference 15 in the footnotes of Table 1, specifying that the table was freely adapted from the article by Askenazi and Perlman. 

--------------------------

Point 2: L79-82, Table 2L162-164: Five cases with congenital anomalies were included in the control cases. Since autopsy was used, congenital anomalies can be a confounding factor in SIUDS and should be excluded. Cases that do not include congenital components, such as cases that died of apparent perinatal problems, should be used as control cases.

Response 2: We believe that the Reviewer is referring to the 5 control cases with hypoplasia of the arcuate and the raphe obscurus nuclei indicated in Table 2. These are slight alterations of the medulla oblongata of little importance,  arising during the fetal development. They cannot be considered a cause of death, as indicated in L158 “are defects that are mostly deemed to be compatible with life (16).”  This clarification was added in the Legends of Table 2.

--------------------------

Point 3:  There is a problem with the description of statistical methods. The method section (L125-128) clearly states only ANOVA, but Tables 3 and 4 are statistical studies of proportions. It is necessary to specify the test method in the footnotes of Tables 3 and 4, rather than just stating that SPSS was used.

Response 3: We have added in the footnotes of Tables 3 and 4 “The Table shows the distribution of the sample proportions”.

--------------------------

Point 4:  In Table 4, it is stated that there was a significant difference only in the top row, 24 with lung hypoplasia (**), but why is there no significant difference in the bottom row, 10-0, and the bottom row, 0-5? Since a 0 in the ratio calculation usually always results in a significant difference, it is necessary to clearly state this in the paper together with the statistical method in 3), and the reason why no significant difference was found should be stated in the discussion.

Response 4: Thanks for this observation. We just wanted to highlight that the most frequent alteration observed in SIUDS with pulmonary hypoplasia, ie the pFn hypoplasia, is present in 23 out of 24 cases associated with maternal smoking in pregnancy. It is also true that the association of maternal smoking with hypoplasia of the pre-Bötzinger nucleus, although less frequent, in this group of SIUDS, is statistically significant. On the other hand, a negative correlation with smoking was found for the hypoplasia of the arcuate nucleus in the 5 control cases. We have modified Table 4 in accordance with all these comments.

--------------------------

Point 5:  Limitation is not mentioned in the discussion, but is it not a limitation that the number and amount of smoking were not investigated (or at least not mentioned in the patherefore this association is highly significantper)? It is hard to believe that the risk of smoking even one cigarette during the entire gestational period is equivalent to smoking more during the entire gestational period.

Response 5: This was our own inaccuracy. We have not indicated in the text that we have defined “smokers” only mothers who have smoked during their entire pregnancy (sometimes even before the beginning), consuming more than three cigarettes a day.

We have thus integrated the text from line 166 onwards: “Overall, 34 of the 47 SIUDS cases (72%) and 4 of the 20 control cases (20%) were exposed to tobacco smoke during the whole pregnancy and often even before the onset of pregnancy, consuming more than 3 cigarettes/day, based on maternal self-reported smoking or cotinine tests.”

Reviewer 2 Report

Sudden intrauterine unexpected fetal death syndrome (SIUDS) (similar to SIDS) represents facets of a multifactorial problem that has not yet found, therefore it is importan to search the potencial etiological factor.  The findins provide new insight into the association between exposure to cigarettes smoking during pregnancy and stillbirth. What is also imortant to notice, that authors not only analyze the effect of active smoking but also include electronic cigarettes.

I do however some concerns:

  1. In my opinion authors should introduce the characteristic of pregnant women (e.g. age, BMI value, race, social status, drug and/or alcohol abuse, etc.) as well as characteristics of stillbirth (fetal weight and the week of pregnancy lose). Especially, when authors in verse 173 wrote „no significant differences were observed regarding the other risk factors such as gender, ethnicity, maternal overweight, drug and/or alcohol abuse”
  2. In verse 94 please introduce the information about the type of samples taken at routine autopsy
  3. Please unify the numeration of Results section: verse 133 and 146 have the same numer a)
  4. Please delete too many spaces between words in whole text
  5. In my opinio it will be more clear to use the number of groups in the table 3 and 4: group 1, group 2 and group 3 not the description of the groups
  6. Please correct „O2” into „O2” (verse 233)

Author Response

Point 1: In my opinion authors should introduce the characteristics of pregnant women (e.g. age, BMI value, race, social status, drug and/or alcohol abuse, etc.) as well as characteristics of stillbirth (fetal weight and the week of pregnancy lose). Especially, when authors in verse 173 wrote “no significant differences were observed regarding the other risk factors such as gender, ethnicity, maternal overweight, drug and/or alcohol abuse”

Response 1:  We have added in Table 3 the data related to the age and socioeconomic status of the mothers and, as regards the fetuses, the data on weight in relation to gestational age.

In Table 3 we had already indicated the values about the presence or absence of maternal obesity but unfortunately we are not able to give the BMI values, since we do not have the data on the height of the mothers. Data on race, drug and/or alcohol abuse were already reported in the Table. Accordingly, we have thus modified the sentence on line 173: No significant differences were observed regarding the other risk factors such as gender, ethnicity, maternal age, overweight, socioeconomic status, drug and/or alcohol abuse.

--------------------------

Point 2: In verse 94 please introduce the information about the type of samples taken at routine autopsy

Response 2: we have thus specified in the sentence on line 94: 

“Samples of major organs (heart, lungs, liver, kidneys, and brain) taken at routine autopsy were fixed in 10% phosphate-buffered formalin, processed and embedded in paraffin. Five μm-thick sections were cut and stained with hematoxylin–eosin and Heidenhain’s trichrome (Azan).”

--------------------------

Point 3: Please unify the numeration of Results section: verse 133 and 146 have the same number a)

Response 3: thanks for this observation. The error has been corrected.

--------------------------

Point 4: Please delete too many spaces between words in whole text

Response 4: this has been made.

--------------------------

Point 5: In my opinion it will be more clear to use the number of groups in the table 3 and 4: group 1, group 2 and group 3 not the description of the groups

Response 5: if possible, we would like to keep this modality in the Tables 3 and 4 as more immediate to understand, without having to report the explanation of the three groups in the legends of the Tables.

--------------------------

Point 6:  Please correct “O2” into “O2” (verse 233)

Response 6:  this has been made.

--------------------------